# Autonomous Crop Row Guidance Using Adaptive Multi-ROI in Strawberry Fields

**DOI:** 10.3390/s20185249

**Published:** 2020-09-14

**Authors:** Vignesh Raja Ponnambalam, Marianne Bakken, Richard J. D. Moore, Jon Glenn Omholt Gjevestad, Pål Johan From

**Affiliations:** 1Faculty of Science and Technology, Norwegian University of Life Sciences, 1430 Ås, Norway; marianne.bakken@sintef.no (M.B.); jon.glenn.gjevestad@nmbu.no (J.G.O.G.); pal.johan.from@nmbu.no (P.J.F.); 2SINTEF Digital, Forskningsveien 1, 0373 Oslo, Norway; richard.moore@sintef.no

**Keywords:** navigation and guidance, image processing, deep learning, automation and robotics

## Abstract

Automated robotic platforms are an important part of precision agriculture solutions for sustainable food production. Agri-robots require robust and accurate guidance systems in order to navigate between crops and to and from their base station. Onboard sensors such as machine vision cameras offer a flexible guidance alternative to more expensive solutions for structured environments such as scanning lidar or RTK-GNSS. The main challenges for visual crop row guidance are the dramatic differences in appearance of crops between farms and throughout the season and the variations in crop spacing and contours of the crop rows. Here we present a visual guidance pipeline for an agri-robot operating in strawberry fields in Norway that is based on semantic segmentation with a convolution neural network (CNN) to segment input RGB images into crop and not-crop (i.e., drivable terrain) regions. To handle the uneven contours of crop rows in Norway’s hilly agricultural regions, we develop a new adaptive multi-ROI method for fitting trajectories to the drivable regions. We test our approach in open-loop trials with a real agri-robot operating in the field and show that our approach compares favourably to other traditional guidance approaches.

## 1. Introduction

Automating agricultural practices through the use of robots (i.e., agri-robots) is a key strategy for improving farm productivity and achieving sustainable food production to meet the needs of future generations. For example, the “Thorvald II” robot [1], developed in Norway, is able to autonomously perform actions such as picking strawberries [2] and applying UV light treatment to greenhouse crops such as cucumbers [3]. A basic requirement for such robots is to be able to navigate autonomously to and from their base station and along the crop rows.

In open fields, real-time kinematic (RTK) GNSS provides an accurate position for the robot but does not inherently describe the location or extent of the crops. Onboard sensors such as scanning lasers [4] or machine vision cameras [5] can enable the robot to sense the crops and structures surrounding the robot directly. However, lidar-based methods work best in structured environments such as greenhouses, and traditional visual approaches rely on distinct and regular crop rows [6] and often employ image features specially crafted for a particular crop appearance.

Norwegian strawberry farms are often located in steep or hilly areas, with crop rows that are not necessarily straight or evenly spaced. Coupled with the dramatically changing appearance of the crop throughout the season (Figure 1) this presents a challenging case for guidance of autonomous agri-robots. We sought, therefore, methods that would enable our Thorvald platform to detect crop rows of varying appearance and to steer the robot accurately along curved or uneven rows.

Convolutional neural networks (CNNs) use many more image features than hand-crafted approaches, and semantic segmentation CNNs [7], trained on per-pixel annotated images, have been shown to be able to distinguish crops from lanes in a more generalized manner, e.g., for identifying crop rows in tea plantations [8]. We similarly adopt a fully-convolutional CNN trained on our own image dataset for detecting strawberry crop rows and show that this approach gives robust segmentations even in difficult scenes.

Extracting agri-robot steering commands from a segmented or thresholded image is traditionally performed by applying a Hough transform [9] or linear regression [10] to leverage the regularity of typical agricultural scenes. However, such approaches will fail when presented with curved or uneven crop rows. Thus, we base our guidance algorithm on the multi-ROI trajectory estimation approach of [11], in which the segmented crop rows are fitted in several smaller ROIs before combining into a single trajectory. However, we find the fixed-width ROIs in this approach less suited to hilly scenarios with varying crop row widths. Therefore, we extend this approach and present a new adaptive multi-ROI trajectory estimation technique that uses a search-based scheme to automatically adapt the ROIs to the varying width of the crop rows.

In this paper, we present a description of our adaptive multi-ROI trajectory estimation approach (Section 3.2 and Section 3.3), as well as results from open-loop experiments with the Thorvald platform (Section 4) that demonstrate the potential benefits of employing this approach for agri-robot guidance in hilly strawberry fields. We also evaluate our approach against other trajectory estimation approaches (Section 5). The main contribution of this paper is the novel adaptive multi-ROI algorithm; however, for completeness we also describe our semantic segmentation approach (Section 3), used to generate segmented input images for the trajectory estimation. We conclude by summarizing our contributions and discussing future work towards closed-loop field testing on board Thorvald (Section 6).

## 2. Related Work

Here, we introduce the traditional visual methods for crop detection and their limitations in various situations. Besides, the crop row fitting using existing methods are reviewed.

Excess Green Pixel Index (EGPI) [12] is one of the widely used techniques in separating vegetation from soil taking into account that only crops have excessive greenness in the image. The index is written in the form:(1)EGPI=2g−r−b
where *r*, *g* and *b* were the color channels of the RGB image. Similarly other color vegetation indices [13] like normalized difference vegetation index (NDI) and the Normalized Green–Red Difference Index (NGRDI) have also been proposed as extensions in segmenting more visually demanding vegetations. In some use cases (Figure 1), the index will begin overestimates the vegetation with the pixels from noncrop regions. Furthermore, it performs poorly if the color of the crop shifts to red or yellow (Figure 1) during late September at the end of the cropping season. For overcoming the reliance of EGPI, Support Vector Machines (SVMs) are introduced in [14], which takes into account the spectral components of the plants. Another prominent work on crop detection is using 3D stereo vision methods [15] in which there is a significant crop and soil height difference, but it will not be the case of early crops. It is visually demanding to differentiate the crop and noncrop region on the inclined fields only based on height. The problem of tiny plants addressed in [16] and applied Dual Hough transform relative to the crop row pattern. [17] uses dynamic programming to generate a template using geometric structures of the crop rows and able to detect the straight and curved crop rows.

Nowadays, segmentation using machine learning methods garners significant interest in precision agriculture. The CNN-based semantic segmentation is used on sugar beet plants [18], remote sensing data of few crop types in agriculture [19] and on rice paddy fields [20]. The performance of CNN-based approaches for detecting crop rows at various stages of growth tested against conventional crop detection techniques [21] and comparatively, the deep learning technique achieves better segmentation results. However, not much work has been done using CNN semantic segmentation in steering the agricultural robots along with the variety of crop rows.

In urban scenarios, the discontinuous lanes are handled by sliding window algorithm [22] where the search windows slide in increments for fitting the driving regions. Similarly, in agricultural cases, blobs [23], or clusters of pixels are used for segmenting the crop regions. Similarly, ref. [24] proposed the concept of multi-ROIs on crop row segmentation that describes the center points estimation of the crops given the inter-row spacing beforehand and applies linear regression over the predicted center points.

However, most of the existing detection methods do not seem to adapt to the crop rows of varying inter-row spacing, which also changes in color, shape, and width throughout the cropping season. The application of multi-ROIs is useful for fitting the varying and curved crop rows. The methodology proposed in this work uses the CNN-based semantic segmentation (SegNet) to identify crops and lanes, which involves the newly annotated dataset generation from the strawberry fields. Moreover, we demonstrate the improved version of the existing multi-ROI algorithm [11] that can adapt to the differing crop rows using incremental search along the equally spaced horizontal strips. Lastly, a quadratic polynomial is applied over center points to obtain the desired centerline (trajectory) for the robot guidance.

## 3. Approach

When the RGB image from the robot mounted camera is received, the main objective of this work is to generate the centerline of the crop on which the agricultural robot is currently positioned. Then the robot can autonomously steer along the predicted centerline till the end of the crop row. The overall approach, as shown in Figure 2 includes the following steps:*Offline CNN Training and Inference*: Annotate the selected RGB images from the data recording in the real fields and do the training with the labeled dataset by semantic segmentation. The well known SegNet has chosen as a base model and gets the weights as a preprocessing step. Predict the incoming RGB image by trained weights and outputs the mask during the CNN inference.*Crop Row Fitting*: With the crop or lane label as a predicted mask, identify the starting points as peaks. Then, the image is equally divided into N horizontal strips in which the adaptive multi-ROI algorithm is applied to find the label centers for each ROIs. Generate the centerline by applying the regression fitting over the estimated label centers.

### 3.1. Data Collection and Annotation

The “Thorvald” platform from Saga Robotics (Agricultural Robotics Company, Ås, Norway) is used for gathering the dataset. The robot is mounted (Figure 1) with the RGB module and a Stereo Camera named Intel Realsense camera D435 (1920 × 1080 max resolution, Rolling Shutter, 69.4 HFOV, 42.5 VFOV, Intel Corporation, United States). The camera is attached using mounting accessories at the height of 1.05 m and 0.785 m in driving direction from the center of the robot. The camera publishes RGB images of size 640 × 480 at the rate of 6 fps. Since the semantic segmentation requires a set of data covering the crops’ diversity, we made a dataset collection campaign on different periods in the strawberry fields with a group of sensors mounted on the robot. The data is recorded in the form of rosbag files provided by ROS. The images are gathered with the RGB camera along with depth information obtained from stereo cameras and can be accessed by the ROS image topic in the respective bag file. The annotations are done by the open-source annotation tool Labelme [25], which gives the annotated images with the label values for each associated pixel. Figure 3 displays the overlay of annotated mask over the corresponding RGB image in which the labeled classes like Background, Crops, and Lanes are stored as 0, 1 and 2 values in the image.

### 3.2. Crop Row Segmentation Network

For semantic segmentation, we applied the well-tested SegNet [26], which is a CNN with an encoder-decoder architecture. We used the implementation with ResNet50 as the base model in the Image Segmentation Keras Library [27], with input size 640 × 352, output size 640 × 480, and 3 output classes. The model is trained on our dataset, using the train/test split described in Section 4. Training is performed with default parameters from [27] for 21 epochs on the Google Colab platform. The adadelta is used as optimizer, batch size is set to two and loss chosen to categorical crossentropy. There is no explicit regularization or augmentation of the dataset. Though the dataset is trained for 21 epochs, the predictions ran with the “early stopping” criterion of 11 epochs due to its lower loss value. Once the training is done, the weights are stored and used to predict the incoming images. The output of CNN is a per-pixel classification with the labels of interest, i.e., crops, lanes, and background in this work.

### 3.3. Crop Row Fitting Using Multi-Roi Algorithm

Once the CNN inference process has completed, the segmented grayscale image is available for estimating the centerline of the crop rows. The next step proceeds with the crop row fitting process by doing the following steps: (a) Identification of starting points as peaks to fit the first ROI, i.e., a rectangular window pointing to the regions of interest, in the label area (b) extract the pixel points belonging to the segmented label using multiple ROIs (c) Adaptive search scheme that automatically adjusts the fitted ROIs to the crop rows of varying width. In the end, the regression fitting is applied over the selected pixel points to determine the robot’s guided line.

#### 3.3.1. Identification of Starting Points

For exploring the pixels of interest, the bottom thirty percent of the segmented image is cropped (Figure 4b,c) and warped to the overhead view for identifying the starting points of the predicted label. The image warping is done using perspective transform *M* that contains the 3 × 3 matrix. The warped image $Img_{W}$ has been obtained using the following:(2)$$Img_{W}=M*Img$$
where *M* is obtained using OpenCV function “getPerspectiveTransform” after the setting the size of warped image $Img_{W}$ (size is chosen as 640 × 640 for optimal estimation). The peaks ${\hat{p}}_{i}$ are estimated by summing the columns of the warped image and obtain the column pixel with maximum white pixels (red/pink colored in Figure 4). For crop-based guidance, at least one peak (Figure 4d) has to be found, whereas the lane-based guidance requires minimum detection of two peaks (Figure 4f) for robot guidance. The estimated peaks are projected back into actual segmented image (Figure 4e,g) using the inverse of the transform matrix *M* and apply inverse mapping over the warped image.

The implementation of the peak estimation used is from the SciPy library with the “find_peaks” function and project the location of the peaks by inverse warping to the original image. When the peaks estimation function could not find any peaks either due to the presence of bare patches or due to camera projection in the hilly terrain, the cropping ratio will be increased to do more exploration. If the minimum numbers of peaks cannot be detected after exploring half of the image, then the current image frame will be rejected for further processing and skips to the next frame.

#### 3.3.2. Extraction of Label Points

As a next step, it was experimentally determined that taking the bottom seventy percent of the image (Figure 4e,g) will give satisfactory results avoiding the overlap of the labels towards the vanishing point. Hence the bottom seventy percent of the image has been cropped to extract the segmented label pixels as label points. At first, the image is equally divided into N number of horizontal strips totaling 10 in this case. The segmented labels have explored from the starting points that are found in the previous step. Each starting point has an independent exploration process that occurs sequentially. A rectangle shaped micro-ROI with fixed left $m_{l}$ and right margin $m_{r}$ is used for identifying the label points beginning from the associated starting point. The height of each micro-ROI is the same as the corresponding substrip (33 pixels). The width varies for each substrip initially set to 100 pixels and decreases gradually at a rate of 5% as it progresses over the substrips. The first micro-ROI have fitted over the projected starting points, as in Figure 5a. The white pixels within the micro-ROI boundary region has taken as label points, and label center $r_{n\in N}$ of the label points within the micro-ROI bounds in the form:(3)$$r_{n\in N}=\left\{x_{cn\in N},y_{cn\in N}\right\}=\left\{\frac{1}{L}\sum_{i}x_{i},\frac{1}{L}\sum_{i}y_{i}\right\}$$
where $x_{i}$ and $y_{i}$ represent the white pixels belonging to the segmented plants inside the micro-ROI and L is the number of white pixels. The first micro-ROI will slide as per the mean of the white pixels within micro-ROI (green rectangle and red circle indicate the micro-ROI and its center in Figure 5a).

The second micro-ROI is applied in the upper substrip (Figure 5b) assigning the label center $r_{2}$ equivalent to the label center $r_{1}$ of the points from the previous micro-ROI. The second micro-ROI will slide based on the mean value of the label points along the second substrip and update the label center $r_{2}$ (blue color in Figure 5c) so that the micro-ROI can contain most of the label points in that particular substrip. The same procedure of applying the micro-ROIs has implemented to the remaining eight strips. Hence the updated label centers $r_{n\in N}$ of the micro-ROI in each substrip (Figure 5d) have obtained. The procedure repeats for other available starting points. Finally, the average label centers are taken as a mean of all the label centers (Figure 5e) belonging to every starting point, and regression fitting is carried out to estimate the polynomial over average label centers as centerline (Figure 5f) for robot guidance.

However, due to its size or shape of the plants, or the hilly terrains in the strawberry fields (Figure 6), the fixed margins of the micro-ROIs could not be able to fit the crop rows of different width. A regression fitting over incorrectly fitted ROIs gives an inaccurate crop row guidance for the robot. The adaptive multi-ROI is proposed in which a search-based scheme uses two sub-ROIs for updating margins of the micro-ROIs, thereby taking the crop rows structure in every respective substrip into consideration.

#### 3.3.3. Adaptive Search Scheme

If the *n*th micro-ROI have at least one white pixel of the segmented label, then the proposed adaptive multi-ROI is introduced to explore the crop rows along the strip. After applying this scheme, the centers $r_{cn\in N}$ of the label points and the respective ROI is updated accordingly. The adaptive scheme has the following steps:To begin the exploration, the left and right sub-ROI (blue and yellow color in Figure 7a) with rectangle shape has initiated from the known label centers $r_{n\in N}$ for every projected peak. The height $\left(h_{rl},h_{rr}\right)$ for the left and right sub-ROI will be constant and equal to the height of the corresponding substrip. The width of both the sub-ROI can be chosen as value based on the crop type to have an optimal exploration area. For the strawberry crops, the size of the sub-ROIs is set to 40 pixels.The sub-ROI on the left (blue) and right side (yellow) from the label centers $r_{cn\in N}$ are constrained to move only in the left and right directions along the corresponding substrip.As in Figure 7b, both the sub-ROI will move in the opposite direction with the constant increment *i*, i.e., the center $c_{l}$ or $c_{r}$ of the left and right sub-ROI will shift *i* number of pixels in their search direction. The sub-ROI will give the flexibility to search along the strip in a sequence until it found the edge of the label or until it reaches the boundaries of the *n*th substrip. A search has also stopped if the current sub-ROI overlap with the already fitted sub-ROI from the neighboring areas.For every increment, a rectangular mask template $mask_{n}$ is generated using the sub-ROI (Figure 8b) in the shape of the respective strip. A bitwise AND operation is performed between the generated template and the *n*th strip $strip_{n}$ (Figure 8a) as:
(4)$$\begin{array}{cc} res_{n} & =strip_{n}\wedge \;mask_{n} \end{array}$$
where the resultant image $res_{n}$ (Figure 8c) will contain the pixels belonging to the label region. The percentage of white pixels in the clusters has calculated, and a check is done if it is less than the minimum threshold *T*. If the threshold is satisfied, then the edge of the label is reached.If the minimum threshold *T* is not satisfied for *i*th increment, then the search scheme repeats from step 3 with the 2ith increment as displayed in Figure 7c. The sub-ROI based search will continue with xi increment (where x=1,2,⋯,16) until one of the stopping criteria is met.When one of the search stopping criteria is met, the distance between the center $r_{n\in N}$ and $c_{l}$ or $c_{r}$ have been updated as new margin values $m_{l}$ or $m_{r}$. The mean of the white pixels within updated ROI bounds has taken as the new label centers $r_{n\in N}$ for *n*th substrip and repeat the entire process for the next substrip until the last substrip is searched (Figure 8d).

Unlike the standard multi-ROI approach, the proposed adaptive multi-ROI based algorithm can dynamically change its path if a row is slanted (Figure 9a) or tiny plants (Figure 9b) or not evenly spaced (Figure 9c). This adaption helps the robot to attain better trajectory generation in visually changing strawberry fields. After the adaptive search scheme, the average label centers have been obtained for regression fitting to estimate the guided line. Algorithm 1 gives an overview of the overall crop row fitting process.
**Algorithm 1** Multi-ROI based Crop Row Fitting.1:**procedure**Multi-ROI Fitting(image)                                ▹ Input2:    **Input:** predicted image, margin $m_{l}$, $m_{r}$;3:    Identification of Peaks ${\hat{p}}_{i}$;4:    **for**
*each peak*
**do**5:        **for**
*each horizontal strip*
**do**6:           Fit ROI with center $r_{n}$ of label points, $m_{l}$, $m_{r}$;7:           Recenter $r_{n}$ as mean of white pixels;8:           **if**
*label points not empty*
**then**9:               Initialize left and right sub-ROIs $R_{n\in (l,r)}$;10:               $m_{l}$, $m_{r}$ = *incremental_search*($R_{n},strip_{n}$);11:               Update ROI with center $r_{n}$;12:        Apply regression fitting over $r_{n\in N}$;13:    Generate guided line $L_{c}$;                                                ▹ Output

## 4. Experimental Setup

Four crop rows are chosen for experiments based on their color variations, different crop width, irregular spacing and projection of their shadows. The RGB images from the selected crop rows are carefully split into training and testing data. Overall, the dataset generalized the variations for training the convolutional neural networks. The selected crop rows have different crop row dimensions after measuring the beginning of the row by hand-held tape. The total image frames for every crop row during the dataset collection are also noted, but the frames involving the end of the crop row are disregarded since they are beyond the scope of this work. The dataset collected for each crop row at different periods has given a naming convention like *YYMMDD_L(N)_(D)* in which the first section mentions the date of data collection in year month date format. It is followed by lane number varying from 1 to 4 for the selected crop rows; otherwise, it uses R for a random lane. Lastly, the direction of the real field is included in which the lane can be faced north or south side.

### 4.1. Dataset

The recorded images were carefully split into separate sets for training and testing of the segmentation network. To achieve a representative training set covering the variation of all the rows, without geographically overlapping with the test set, the following splitting procedure was followed: (a) Every dataset was divided into three parts. (b) Twenty chosen images from two parts are taken for training, and ten images from the remaining part are kept for testing. (c) In the case of dataset facing north or facing south, twenty images from two parts in one direction and ten images from another direction are taken. The labeled dataset consists of 317 images for training and 120 for testing in total. The camera images and the corresponding annotations are made publicly available https://doi.org/10.6084/m9.figshare.12719693.v1.

### 4.2. Crop Row Segmentation

To visualize the quality of the segmentation, per-pixel results of the lane class with its true positives, undetected vegetation and undetected soil are plotted on top of the input image. Quantitative assessments of the segmentation algorithm in isolation are outside the scope of this paper.

### 4.3. Adaptive Multi-Roi Experiments

The proposed adaptive multi-ROI is a subset of the actual multi-ROI algorithm. The sub-ROIs will begin incremental search only if there is at least one white pixel within the ROI. Otherwise, it will be termed as seasonal plants and skip to the next horizontal strip or complete the fitting procedure. The generated label centers $r_{n}$ are given to regression fitting for fitting either the straight line or the polynomial based on the residual. By comparing the fitted image with the output image from the segmentation predictions, the multi-ROI fitting procedure is evaluated. The performance of various crop row fitting methods is shown in terms of intersection over union (IoU) (Equation (Equation 5)) as:(5)$$\begin{array}{cc} Fitting\;IoU & =\frac{mask\cap mask_{fit}}{mask\cup mask_{fit}} \end{array}$$
where mask is the output image from CNN segmentation taken as ground truth and $mask_{fit}$ is the template generated by fitting methods. Moreover, the metric called “Crop Row Detection Accuracy” (CRDA) introduced in [17] will be applied to check how well the estimated centerline is to the ground truth images. The performance metric CRDA is computed by using the horizontal coordinates $x_{c}$ of the centerline $L_{c}$ for each crop row to the corresponding horizontal coordinates ${\bar{x}}_{c}$ of the ground truth images. The equation is written with the number of the crop rows (*N* = 1) and the number of image rows (*M* = 252) as
(6)$$\begin{array}{c} CRDA=\frac{1}{N\times M}\sum_{i=1}^{M}\sum_{j=1}^{N}S^{*}\left({\bar{x}}_{c,i},x_{c,i},s_{c,i}\right) \end{array}$$
where
(7)$$\begin{array}{c} S^{*}\left({\bar{x}}_{c},x_{c},s_{c}\right)=max{(1-\left(\frac{{\bar{x}}_{c}-x_{c}}{\eta s_{c}}\right)}^{2},0) \end{array}$$

However, only one crop row is detected in these use cases, the matching score equation ($S^{*}$) is normalized with the known crop row spacing $s_{c}$ in pixels instead of inter-row spacing parameter *s* [11] in Equation (Equation 7). However, the crop rows in the test sets have varying spacing, therefore max($s_{c}$) has been used with the scaling factor η for each horizontal strip. The average matching scores belonging to all the image rows (*M*) will give a metric representation within the range of 0 to 1 as the worst and best estimates.

#### Ground Truth Images

For evaluation, each image for testing should have the centerline as ground truth. Since each image set is different, the ground truth has to be generated by manual annotation (Figure 10). The plants are more extensive so that the crop boundaries are annotated manually as a polygonal ROI, and a binary mask is generated. The image is divided into *N* strips. A center point ${\bar{x}}_{c}$ is the average between the min and max *x* values for each substrip of the center crop mask. In this way, every substrip has center points and runs a regression fitting over all the center points in *N* strips. These ground truth values are associated with each image used by CRDA metrics for evaluating the estimated centerline by four crop fitting methods. In total, 400 ground truth images have been created.

## 5. Results

### 5.1. Crop Row Segmentation

Figure 11 shows some example segmentation results for the lane class, to provide some intuition about the quality of the segmentation. Here, the quality of the segmentation is overall very good, also for difficult cases with in-line vegetation and shadows. Most of the false positives and negatives are due to inaccurate labels.

### 5.2. Crop Row Fitting with Adaptive Multi-Roi

The usage of semantic segmentation gives the flexibility of using the labels of interest. The crop row fitting using adaptive multi-ROI can be tested either on the crop or lane labels since both can generate the guided centerline for the robot. The methodology has been designed as such the guided line can be estimated as an average of $L_{n}$ odd number of fitted crop labels or $L_{n}$ even number of fitted lane labels. For evaluating the various use cases, the image sets have been organized as follows: larger plants, tiny plants, uneven plants and inclined terrains. The images suited for every use case are handpicked from the gathered datasets. The crop row fitting methods have been tested over four image sets, each containing 100 images.

At first, the standard multi-ROI was tested on the image sets for evaluating its performance. The multi-ROI fitting performed over the predicted image for the lane label. It is noticeable that the multi-ROI fitting could not handle the change in width among the crop rows (Figure 12a–d). These led to the incorrect estimation of the centerline, which makes the robot deviate from the driving region. To overcome this problem, the adaptive multi-ROI is introduced, which does an incremental search algorithm along the strip for estimating the ROI margins. The search algorithm finds the exact margins for each ROIs in all the test cases. The predicted labels in Figure 12e–h detail the centerline generation over the estimated centers $r_{n\in N}$ on two of the predicted lanes region. Then, the centerline has been determined by regression fitting with the average points of both lines. Similar tests have been performed with the crop label where only one crop has fitted (Figure 12i–l); therefore, the fitted crop was taken as the centerline.

Table 1 shows the average total execution time and obtained fitting IoU using multi-ROI and adaptive multi-ROI methods with the test sets. The fitting IoU tells how well the crop rows have fitted. The proposed adaptive multi-ROI take more time in execution than the multi-ROI due to the additional incremental search time taken by sub-ROIs. Even though the adaptive multi-ROI consume more time, the fitting IoU is much higher, determining better overlays with the predicted mask. Hence it can be concluded the adaptive multi-ROI perform the fitting better than the standard method in all the use cases.

The performance of the proposed multi-ROI crop row fitting methods have been evaluated along with the traditional line fitting approaches on the image sets. The modified version of the CRDA metric has been used for evaluating the crop row fitting methods. At first, the probabilistic Hough line transform ran on the image sets to find the centerline and evaluate it with the ground truth images. The RGB image has warped into an overhead view, and Hough transforms fit a group of straight lines over predicted labels. The Hough transform parameters have fine-tuned to achieve better results for these use cases, and then the average of all the lines has taken as the guided line. The next method involves the linear regression based on the least-squares that use templates created by areas containing the predicted label. In this method, the contours are found around the labels (crop or lanes) in the image. Then the straight line is fitted over each contours region by minimizing the objective function. Smaller labels that are farther away in the image are removed by their contour area to improve the estimation performance. The centerline can be taken as an average of fitted lines and compared with ground truth. Moreover, the estimated centerline from the multi-ROI and the proposed adaptive multi-ROI methods has also been evaluated using the image sets.

From the tests using crop and lane labels on image sets, the Table 2 displays the mean and standard deviation of the CRDA scores by fitting methods. The Hough transform has decent results (>0.55*) on all image sets but perform poorly when the crop rows are broader, like in the tiny plants (lane label) or larger plants (crop label) set. In linear regression, the results are slightly better (>0.60*) than Hough but give below optimal accuracy when it comes to irregularly shaped crop rows. The multi-ROI technique achieves results close to 0.80* on image sets but suffers on the situations when crop row has a larger width than the ROI margins. It could be improved if the margin of the ROI was preset based on the crop row width, which would not be practical while running online in real fields. The adaptive multi-ROI methods give improved results over standard multi-ROI by dynamically handling the crop rows width. Therefore, the introduction of adaptive multi-ROI crop row fitting gives improved results on guidance with CRDA achieves close to 0.90* in most test cases. The video of the proposed crop row guidance approach tested over four different crop rows is available online: https://youtu.be/IxK51ewD6os.

## 6. Discussion

The performance of the entire crop row guidance depends on the quality of the segmentation and the crop row fitting. The output from the segmentation part has shown good accuracy when used together with the multi-ROI fitting. Further quantitative tests of the segmentation network and its ability to generalize across different rows and seasons are beyond this paper’s scope and will be treated in the future paper. In this paper, the focus is mainly on the multi-ROI fitting. Even though the entire crop row guidance system seems to work well for most scenarios, there are tricky situations where the crop row is reduced to a certain accuracy.

It is common in real fields when the plants do not always grow sequentially along the crop rows, either due to soil compaction or lack of sunlight, which leads to specific gaps termed as bare patches. As in Figure 13a,d, due to the bare patches, the fitting using crop label could handle the gaps in the segmented crop rows and shifts to the next substrip (Figure 13b) for the further fitting process. Thus the crop row fitting pipeline has not been interrupted in these scenarios. However, fitting using lane labels will misfit the label points belonging to one segmented lane to the neighboring lanes. The misfit leads to two situations. First, since the overlapping of micro-ROIs is not permitted, the micro-ROI from the first lane fit the label points from the second lane leaving the micro-ROI from the second lane with no exploration area. For the overlapping case, the situation has been handled by assigning the same center values for the micro-ROIs (Figure 13c) of the overlapping lanes. The second case is when the micro-ROI explore the neighboring lane, which is not a part of the crop row fitting procedure. This scenario generates the deviated guided line like zig-zag movement that does not replicate the actual scenario (cyan in Figure 13f). This situation has been identified by finding the radius of curvature R of the guided line and then set the limit for the value to be greater than a certain minimum value. The radius of curvature can be calculated as
(8)$$R=\frac{{\left(1+{\left(2ax+b\right)}^{2}\right)}^{\frac{3}{2}}}{2a}$$
where *a* and *b* are the polynomial coefficients by fitting the coordinates of the guided line, *x* is the horizontal coordinates of the guided line. If the limit condition is not satisfied, then the most distant center among all center points of the guided line along the horizontal axis is taken as an outlier and the outlier value replaced with the horizontal center from the previous substrip. The procedure is repeated until the limit condition is satisfied and suboptimal guided centerline (pink in Figure 13f) is obtained even in tricky cases.

All the discussed crop row fitting methods have a minimal performance on the inclined terrains, mainly due to the camera visualization of the labels in which some sections of the labels are not visible. Hence it makes the estimated centerline also inclined towards the more visible side. It could be improved by skewing the image based on the inclination information obtained from other sensors like calibrated IMU or depth cameras.

## 7. Conclusions

In this work, we have proposed a crop row guidance system by vision systems using a CNN-based segmentation combined with adaptive multi-ROI based crop fitting. With the evolution of machine learning, the crop models can be learned with the big data and adapted to the changes in the environments. For generating the essential big data required for machine learning, the onboard sensors mounted on the agricultural robot deployed for the data collection. Furthermore, the proposed crop fitting procedure applies multiple smaller windows along the equally divided horizontal strips on the input image. The methodology has estimated the guided line to higher precision in most of the use cases after analyzing with the ground truth.

Furthermore, the tests showcased improved results compared with the other fitting methods. Since the agricultural robot runs at an average of 1.5 km/h, the proposed methodology could run online in the real fields. The future work involves finding the need for the automatic transition when the robot reaches one end of the crop row by following the proposed crop row guidance. The extension will be developing the robot to autonomously maneuver to nearby crop rows and follow the crop row guidance in the next row. Therefore, the agricultural robot will have closed-loop field testing for the entire field.

## Figures and Tables

**Figure 1 sensors-20-05249-f001:**
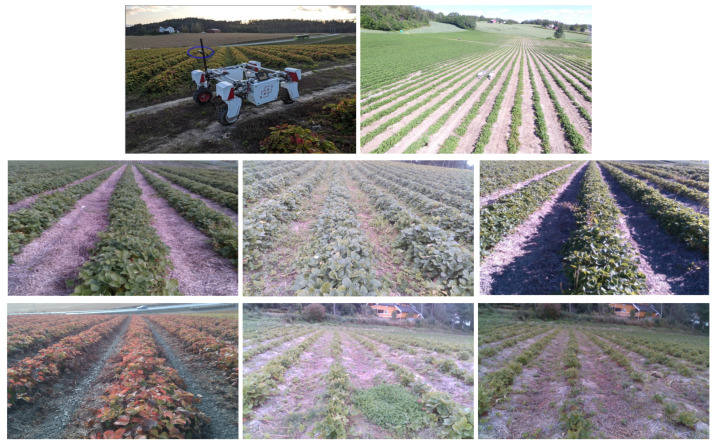
Top: The Sensors mounted on the “Thorvald” Robot Platform for dataset collection on hilly terrains. The variation of crop rows from the strawberry fields in Norway have illustrated from the RGB images listed above. Second Row: The transition of plant growth from July to September in 2019 defines the illumination problems in which the plants or shadow covers the entire driveable areas. Third Row: presents the change in color of the plants throughout the season and showcase the problem of random plants growing over lanes or the occurrence of bare patches along with the crops.

**Figure 2 sensors-20-05249-f002:**
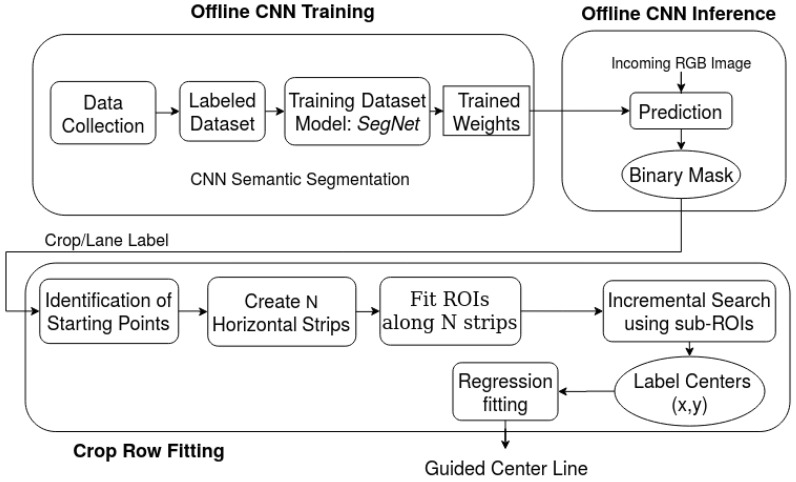
Pipeline for the proposed Adaptive Multi-ROI based crop row guidance is presented as three sub parts: *Offline CNN Training* involves data collection and training the dataset, *Offline CNN Inference* does CNN prediction over incoming RGB image, and *Crop Row Fitting* applies the fitting procedure over predicted mask by the proposed adaptive multi-ROI method.

**Figure 3 sensors-20-05249-f003:**
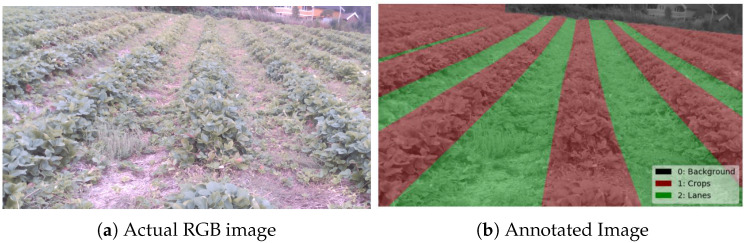
Overlay of the annotated mask on the RGB image for training.

**Figure 4 sensors-20-05249-f004:**
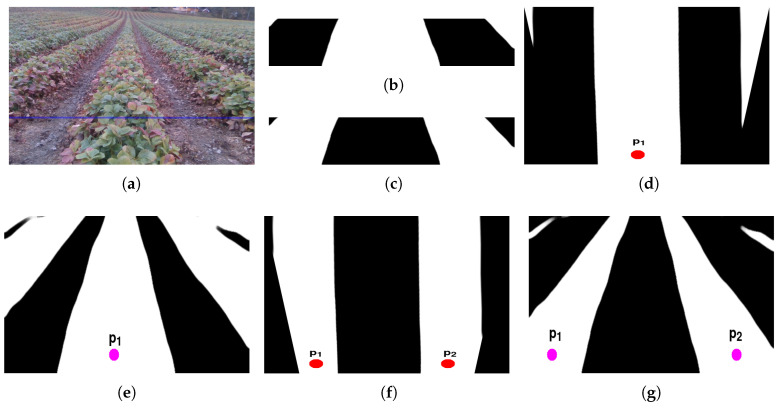
The peaks are estimated (red) on both the crop and lane label by summing the columns of the warped image. The peaks are projected (pink) back by inverse warping on the segmented image. (**a**) Incoming RGB image, (**b**) Crop Label, (**c**) Lane Label, (**d**) Estimated Peaks (Warped Lane Label), (**e**) Projected Peaks (Inverse Warped Lane Label), (**f**) Estimated Peaks (Warped Crop Label), (**g**) Projected Peaks (Inverse Warped Crop Label).

**Figure 5 sensors-20-05249-f005:**
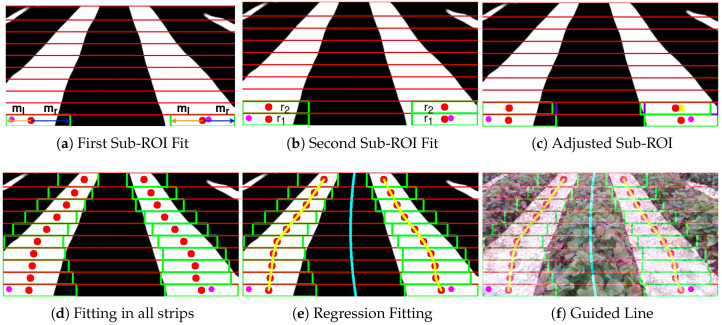
Step by Step Procedure for fitting multiple windows over the pixels of interest as (**a**) Window applies over peaks and adjusted based on mean values of the white pixels. (**b**) The neighboring strip has a second window with the same center as the previous window. (**c**) Second window adjusted with the mean of white pixels within its boundaries. (**d**) The same method extends until the last horizontal strip. (**e**,**f**) Estimating the centerline for crop row guidance using regression analysis.

**Figure 6 sensors-20-05249-f006:**
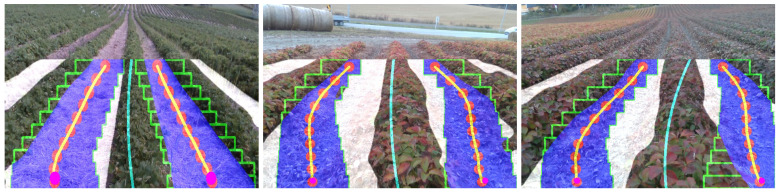
The above scenarios highlight the issue of the multi-ROI fitting using fixed margins $\left(m_{l},m_{r}\right)$ to the crop rows having different width. Courtesy: identified peaks (pink), rectangular boxes (green) indicates the ROIs and the chosen label as label points (blue) within bounds.

**Figure 7 sensors-20-05249-f007:**
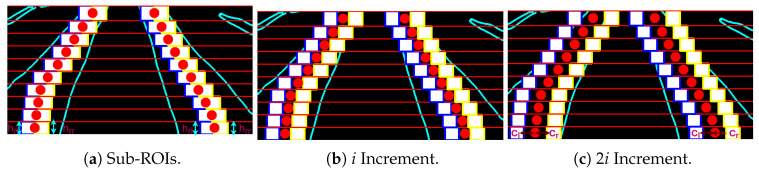
The proposed Adaptive Multi-ROI methodology to adapt to the varying crop row widths using incremental search along horizontal strips.

**Figure 8 sensors-20-05249-f008:**
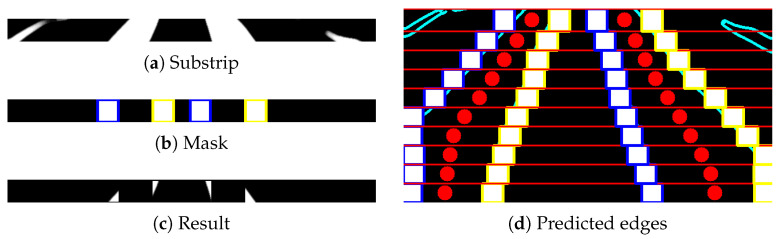
The figure showcases the resultant image obtained using substrip and mask that helps to find the edge of the labels. The sub-ROIs are stopped when it reaches the edges and update the label centers for each micro-ROI.

**Figure 9 sensors-20-05249-f009:**
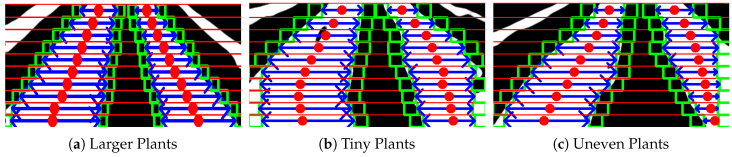
The figure showcases the usage of the proposed adaptive multi-ROI methodology in the cases comparing to the existing multi-ROI method.

**Figure 10 sensors-20-05249-f010:**
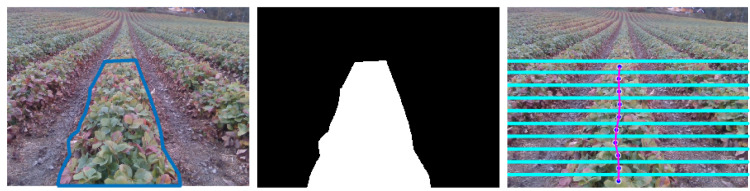
Ground Truth Centerline by manual annotation. The left image is the raw RGB image from the camera. The middle image is the binary mask generated by freehand masking; the final image corresponds to the centerline of the crop row as ground truth (pink).

**Figure 11 sensors-20-05249-f011:**
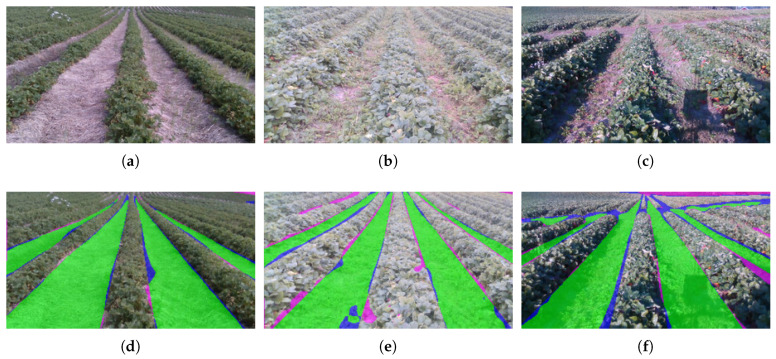
Presents the example input images during different stages of the harvesting season (**a**–**c**) and their CNN segmentation results for the lane class (**d**–**f**). Green is correct prediction, blue is undetected soil, magenta is undetected vegetation. Best viewed in color.

**Figure 12 sensors-20-05249-f012:**
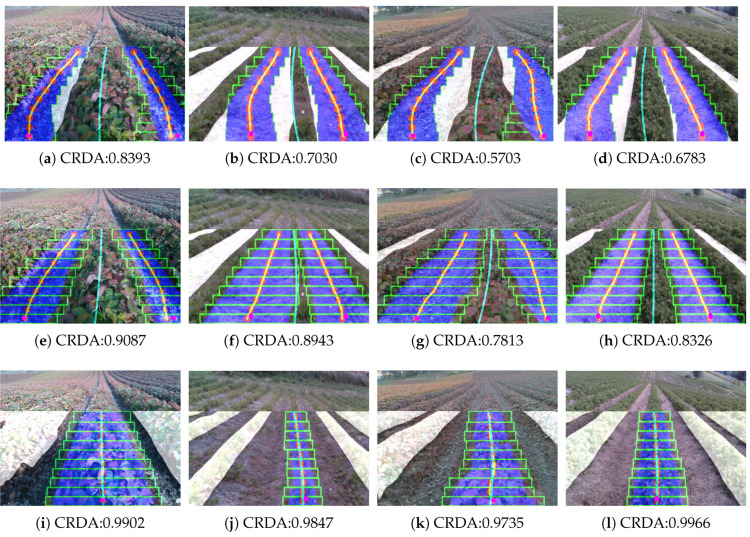
Fitting results for four test cases depicting the variance of strawberry crop rows. Columns represent Multi-ROI fitting results using fixed-size ROIs (green) (**a**–**d**). The proposed Adaptive Multi-ROI fitting results using adaptive-size ROIs (green) are showcased for lanes (**e**–**h**), and crops label (**i**–**l**). The Adaptive Multi-ROI can find the cluster centers (red) in the lane or crop regions. A regression fitting applied over the average of fitted labels (yellow) and obtained the centerline (cyan) for the robot to follow the crop row. The metric CRDA tells how well the centerline for autonomous guidance is estimated against the ground truth.

**Figure 13 sensors-20-05249-f013:**
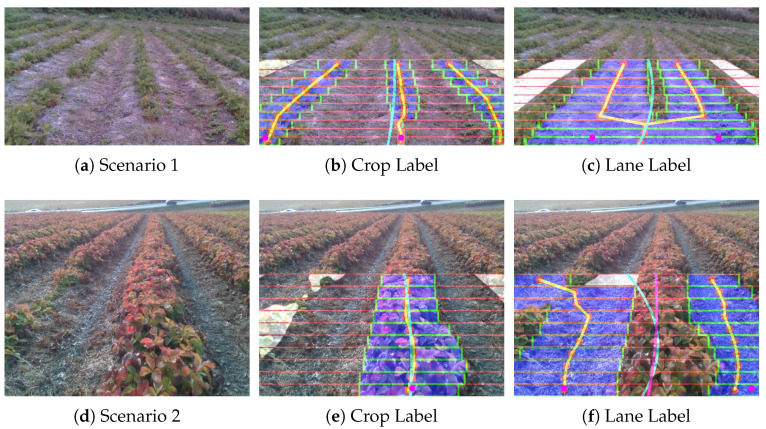
Handling of bare patches scenarios using adaptive multi-ROI fitting.

**Table 1 sensors-20-05249-t001:** Comparisons of fitting IoU between multi-ROI and proposed Adaptive multi-ROI approach.

	Multi-ROI		Adaptive Multi-ROI	
**Test Set**	**Execution**	**Fitting**	**Execution**	**Fitting**
**(100 Images)**	**Times**	**IoU**	**Times**	**IoU**
Larger Plants	0.40 ± 0.01	0.9003	0.68 ± 0.05	0.9615
Tiny Plants	0.21 ± 0.01	0.7328	0.41 ± 0.03	0.8924
Uneven Plants	0.25 ± 0.01	0.8482	0.53 ± 0.01	0.9100
Inclined Terrains	0.30 ± 0.00	0.8365	0.63 ± 0.02	0.9378

**Table 2 sensors-20-05249-t002:** Crop Row Detection Accuracy for the estimated centerline over the testing image sets.

	Hough Transform		Linear Regression		Multi-ROI		Adaptive Multi-ROI	
**Test Set**	**Crops**	**Lanes**	**Crops**	**Lanes**	**Crops**	**Lanes**	**Crops**	**Lanes**
**(100 Images)**	**Label**	**Label**	**Label**	**Label**	**Label**	**Label**	**Label**	**Label**
Larger Plants	0.6450 ± 0.31	0.7221 ± 0.17	0.8754 ± 0.11	0.7134 ± 0.18	0.7155 ± 0.46	0.9251 ± 0.09	0.9599 ± 0.06	0.9457 ± 0.06
Tiny Plants	0.5306 ± 0.34	0.5456 ± 0.15	0.7963 ± 0.22	0.8035 ± 0.18	0.9382 ± 0.11	0.7753 ± 0.07	0.9679 ± 0.06	0.8933 ± 0.04
Uneven Plants	0.6719 ± 0.32	0.5631 ± 0.13	0.8333 ± 0.21	0.5136±0.12	0.9142 ± 0.11	0.8636 ± 0.08	0.9679 ± 0.06	0.9333 ± 0.06
Inclined Plants	0.5085 ± 0.34	0.5050 ± 0.06	0.8183 ± 0.21	0.3324 ± 0.13	0.8857 ± 0.27	0.7929 ± 0.11	0.9835 ± 0.02	0.8801 ± 0.06

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
