# Peer review of "Autonomous Crop Row Guidance Using Adaptive Multi-ROI in Strawberry Fields"

_sensors, 2020, doi:10.3390/s20185249_

Round 1

Reviewer 1 Report

Good paper. Well structured, well written. Very interesting

Author Response

Point 1: EGPI I recommend reporting the function and describing the factors; greenness index. See row 59;

Response 1: The EGPI index is added as Equation 1. Added popular indexes are also mentioned.

Point 2: Saga Robotics - Add specifications, manufacturer city, and state; Intel Realsense .... Add specifications, e.g. max resolution, optics, FOV, manufacturer city and state, etc; Specify better the setup used in the tests and the characteristics of the chamber

Response 2: All the related information about the sensors and company are added now. The setup of the sensor rack for placing the camera is also described in brief.

Point 3: 134-138  The process is not very clear. 30% of 480 pixels = 144 pixels? So the resulting image becomes 640 x 144 px? Figure 4 which is not very explanatory.

Response 3: Sorry for not providing enough clarity on image warping. Now I have added detailed information that describes the step by step procedure pointing to the respective figures.

Point 4: Do the images represent 30% of the image (the crop)? It would be appropriate to add the original image at the beginning of the sequence (highlighting the limits of the crop). Is it not clear how the stretching and reprojection process takes place? What parameters are used? On the left three peaks are identified, why then they become 2, which one discards. It would be advisable to mark (with letters or numbers) the corresponding peaks in the 3 images

Response 4: Now the original RGB image with the limits of the crop is added. The steps of projection and reprojection are explained both visually and in text now.  A perspective transform matrix is used for warping and inverse warping. I have marked the peaks in the letters for better understanding.

Point 5: What is the default height?

Response 5: The default height is 33 pixels. Since 336 in height is equally divided into 10 strips with the exception of the last strip in the top of the image has 36 pixels.

Point 6: Highlight the edges of the segmentation (gray line)

Response 6: The Edges in the Figure 7 and Figure 8 (d) are highlighted in cyan color.

Point 7: 222-223 False negative or positive? I suggest undetected vegetation, undetected soil. Explain better

Response 7: The wordings have been changed as per suggestions in Sec 4.2 and also in Figure 11 labelings.

Point 8: Add (Fig. 10); I suggest: delete quote and complete the sentence, eg.: a future paper

Response 8: All the grammatical errors and other small suggestions have been rectified in the paper.

Reviewer 2 Report

The manuscript was well written to understand easily and the effective approach for image-based crop row detection was presented. The study has provided interesting image processing pipeline for autonomous farming machine and the reader can be understood easily since the contents written logically.

  • Maybe lack of my understanding… I’m not sure what strengths your multi-ROI (also adaptive) has in crop row guidance. The target areas have already been detected through SegNet, and the guide curve could be estimated by using the inner edges of lane areas. In adaptive multi-ROI approach, the left (right) boundary was shifted to meet the inner edge of segmented area. Please elaborate the contents to understand easily.
  • Line 107: Why did you use RGB images for area segmentation? A stereo camera was already equipped in your platform, “Thorvald”. I think that disparity map by stereo matching can provide practical pipeline in field without CNN-based segmentation that required high overhead resources.
  • Line 117: The authors have cited references, but the reader will want to know more detail processing for SegNet training (e.g. stop condition for 21 epochs, optimizer, regularizer, mini-batch)
  • Line 133: It's a good idea to explain more about image warping
  • Line 148: Please fill the all sub-title, (a)~(f)
  • Line 225: lowercase -> Intersection over union
  • Check the title of Figure 12. It is hard to match the columns (a), (b), and (c) with context.
  • Line 172: Some notations were used only in text. Please include the notations in the figure or equation for understanding (e.g. hrl, hrr).

Author Response

Point 1: Maybe lack of my understanding… I’m not sure what strengths your multi-ROI (also adaptive) has in crop row guidance. The target areas have already been detected through SegNet, and the guide curve could be estimated by using the inner edges of lane areas. In adaptive multi-ROI approach, the left (right) boundary was shifted to meet the inner edge of segmented area. Please elaborate the contents to understand easily.

Response 1: Since the structure of the crop rows is uneven, the detection using SegNet alone will not be able to estimate the guided line. This is due to a couple of reasons: (a) CNN cannot guarantee 100 percent accuracy in all cases that will lead to the inaccurate guideline (b) it is common to have gaps in the plants that will be difficult to differentiate using simple vision techniques. (c) the smaller plants affect the estimation of the inner edges. When using multi-ROI, the idea is to explore the interesting pixels and try to stay around the mean of those interesting pixels. Even if predictions are not 100 percent accurate or when there are gaps, the guided line can be estimated to much better accuracy.

Point 2: Line 107: Why did you use RGB images for area segmentation? A stereo camera was already equipped in your platform, “Thorvald”. I think that disparity map by stereo matching can provide practical pipeline in field without CNN-based segmentation that required high overhead resources.

Response 2: That's true. Since the strawberry fields are grown in the inclined terrains, the disparity map can not able to distinguish between crops and lanes in many cases when both of them almost share the same height. Therefore the CNN became a go-to procedure for understanding the variance.

Point 3: Line 117: The authors have cited references, but the reader will want to know more detail processing for SegNet training (e.g. stop condition for 21 epochs, optimizer, regularizer, mini-batch)

Response 3: I have added more details about the SegNet Training explaining the parameters that have been used during the training process.

Point 4: Line 133: It's a good idea to explain more about image warping

Response 4: I have added a step-by-step procedure for the image warping in Section 3.3.1. And also added more figures (Figure 4) for better understanding.

Point 5: Line 148: Please fill the all sub-title, (a)~(f); Line 225: lowercase -> Intersection over union; Check the title of Figure 12. It is hard to match the columns (a), (b), and (c) with context.

Response 5: The subtitles are added and altered for the better understanding as per suggestions.

Point 6: Line 172: Some notations were used only in text. Please include the notations in the figure or equation for understanding (e.g. hrl, hrr).

Response 6: Notations are included in the respective Figures.